# A Soil Moisture Prediction Model, Based on Depth and Water Balance Equation: A Case Study of the Xilingol League Grassland

**DOI:** 10.3390/ijerph20021374

**Published:** 2023-01-12

**Authors:** Rong Fu, Luze Xie, Tao Liu, Binbin Zheng, Yibo Zhang, Shuai Hu

**Affiliations:** 1College of Economics, Hangzhou Dianzi University, Hangzhou 310018, China; 2Department of Sociology, Hangzhou Dianzi University, Hangzhou 310018, China; 3College of Computer Science and Technology, Hangzhou Dianzi University, Hangzhou 310018, China

**Keywords:** soil moisture prediction, water balance equation, seasonal ARIMA model, sustainable development, land use

## Abstract

Soil moisture plays an important role in ecology, hydrology, agriculture and climate change. This study proposes a soil moisture prediction model, based on the depth and water balance equation, which integrates the water balance equation with the seasonal ARIMA model, and introduces the depth parameter to consider the soil moisture at different depths. The experimental results showed that the model proposed in this study was able to provide a higher prediction accuracy for the soil moisture at 40 cm, 100 cm and 200 cm depths, compared to the seasonal ARIMA model. Different models were used for different depths. In this study, the seasonal ARIMA model was used at 10 cm, and the proposed model was used at 40 cm, 100 cm and 200 cm, from which more accurate prediction values could be obtained. The fluctuation of the predicted data has a certain seasonal trend, but the regularity decreases with the increasing depth until the soil moisture is almost independent of the external influence at a 200 cm depth. The accurate prediction of the soil moisture can contribute to the scientific management of the grasslands, thus promoting ecological stability and the sustainable development of the grasslands while rationalizing land use.

## 1. Introduction

As a natural ecohydrological resource, soil moisture (SM) is an important aspect of the heat transfer process and energy exchange between the surface and atmospheric system, as well as the key link of the surface and groundwater circulation and the land carbon cycle [1]. It plays an important role in many aspects. In the ecological field, changes in the soil moisture are associated with changes in the vegetation cover and have a significant impact on the forest decline, land degradation and loss of biodiversity [2]. In the field of hydrology, soil moisture is an important indicator to assess the degree of drought, and different soil moisture levels can bring about deviations in the physical patterns [3]. In agriculture, on the one hand, soil moisture affects the length of the grass growing season, the growth rate of grasses and the nutrient uptake; too much soil moisture affects the crop root respiration and too little soil moisture limits the crop uptake of fertilizers [4]. On the other hand, it limits the dissolution and transfer of nutrients and the microbial activity in the soil [5], leading to the nutrient loss to the wider environment and thus affecting the crop yields. In the climate domain, soil moisture limits transpiration and the photosynthesis of plants, which in turn has an impact on water, energy and biogeochemical cycles, while the persistence of soil moisture leads to persistence of the climate system. Conversely, changes in climate also have an effect on soil moisture, creating a soil moisture-climate interaction [6]. Therefore, soil moisture is of critical importance in ecology [7], hydrology [8], agriculture [9] and climate change [10,11] research, and the accurate prediction of the soil moisture can provide important guidance for drought prevention, grazing irrigation and other scientific applications [12].

Soil moisture is influenced by a variety of factors, such as precipitation [13], evaporation [14], soil properties [15] and vegetation [16], and the predictability of the soil moisture may result from its own persistence or external forcing factors [17]. Thus, the soil moisture prediction exhibits a degree of uncertainty, and the highly nonlinear variation relationship between the explanatory variables and the soil moisture makes its prediction very difficult [18]. Prediction methods for the soil moisture fall into two main categories: physics-based process models [19] and data-driven empirical models [20]. Physics-based process models focus on the hydrological processes that control the soil moisture transfer mechanisms through physical equations, and calculate the explanatory variables as part of the land surface data assimilation techniques [21]. Factors, such as precipitation, atmospheric temperature and solar radiation, driven by model-generated or observationally obtained factors, can be used for the seasonal flow prediction, such as soil moisture, runoff and so on [22]. For example, Huang explored the dynamics of the soil moisture in the Weihe River basin through water balance equations [23]; Tang used a two-parameter monthly water balance model to predict the runoff [24]. However, the physical model is complicated, and the absence or lack of information may bring some errors to the soil moisture prediction. Furthermore, because the equations are based on the assumption of the distinctiveness of the partial rules, they are not applicable to the medium- and long-term predictions. The data-driven empirical models are based mainly on constructing maps of explanatory variables and soil moisture. The widely used statistical methods include multiple linear regression models [25,26], Bayesian models [27] and support vector machine models [28,29]. For example, time series analysis methods are used to build autocorrelation models of the historical soil moisture for predicting the future soil moisture [30]. Multiple regression analysis methods are also used to estimate forecasts by establishing multiple regression equations between the explanatory variables and the soil moisture [31]. Currently, many artificial intelligence methods and techniques also exist for the soil moisture prediction [32,33], and the data sources are mainly satellite remote sensing data and microwave radar data [34]. Satellite remote sensing imaging, although intuitive, is often only applicable to the exploration and monitoring of materials visible on the surface [35]. In addition, it is difficult to predict the soil moisture at sites or areas where data are sparse or unavailable, and at the same time, changes in weather patterns, as well as the persistence and severity of precipitation events, can produce a high randomness in the observed values [36].

In order to accurately predict the soil moisture at different depths in the Xilingol League grassland of Inner Mongolia, and then to provide reference for the research of the grazing ecology theory and sustainable grassland utilization practice, a soil moisture prediction model, based on the depth and water balance equation is proposed to address the shortcomings of the physically based process model and the data-driven empirical model. The model combines a water balance equation and the seasonal ARIMA model. The model predicts the soil moisture at 10 cm, 40 cm, 100 cm and 200 cm in the test plots, by considering the effect of the depth variation on the soil moisture and thus introducing a depth parameter γ. The model has a high prediction accuracy in predicting the soil moisture at 40 cm, 100 cm and 200 cm. Different models were used for different depths. In this study, the seasonal ARIMA model at 10 cm and the proposed model at 40 cm, 100 cm and 200 cm resulted in more accurate prediction values and provided an important reference for developing the appropriate irrigation, grazing plans and water management, as well as improving drought prediction and agricultural crop yields.

## 2. Materials and Methods

### 2.1. Study Area and Data Acquisition

The Xilingol League grassland in Inner Mongolia is located on the Xilin River in the Inner Mongolia Plateau, with geographical coordinates between 110°50′~119°58′ E and 41°30′~46°45′ N. The total area of the Xilingol League grassland is 19.2 million hectares, and the grassland landscape is a high plain with an elevation between 800 and 1200 m. In the climate zone, the Xilin River basin belongs to the temperate subarid region, and its climate type belongs to the temperate semi-arid grassland climate in the continental climate. The four seasons are distinct: spring is often accompanied by windy weather and summer is warm and humid, while autumn and winter are cold and dry. According to the meteorological monitoring data of the Inner Mongolia grassland station, the annual sunshine hours in the area is 2603.8 h. The average annual temperature is 0.96 °C and the potential annual evaporation is about 1664.6 mm. The annual precipitation is about 340 mm, and the inter-month variation of the precipitation is large, mainly concentrated in June-September, accounting for about 74.4% of the annual precipitation. The specific soil type is mainly kastanozem, and the plant community is most widely distributed by the sheep grass (Leymus chinensis) community and big needle grass (Stipa grandis) communities. The Inner Mongolia Xilingol League grassland is not only an important national livestock production base, but also an important green ecological barrier that plays a role in reducing the occurrence of dust storms and severe weather. It is also one of the typical areas for studying the ecosystem response mechanisms to human disturbance and global climate change, as well as an important part of the terrestrial sample belt in the International Geosphere-Biosphere Program (IGBP)—Northeast China Terrestrial Ecosystem Sample Belt (NECT). Therefore, the accurate prediction of the soil moisture has a positive impact on the sustainable development of the grasslands.

The geographic location of Xilingol League is shown in Figure 1. The observation point for this study was 115°22.5′ E, 44°7.5′ N and the soil type studied was kastanozem. The soil is classified as kastanozem, based on the World Reference Base for Soil Resources, with 37% clay content (<0.002 mm), 42% silt content (0.002–0.02 mm), and 21% sand content (0.02–2 mm). The profile construct is the Ah-Bk-C type. The organic matter content of this soil is 1.2%, the soil pH—7.3, the cation exchange capacity—22 cmol/kg, the base saturation percent—100%, the calcium carbonate content—6%. The variables used were precipitation (P), soil evaporation (*SE*), normalized difference vegetation index (*NDVI*), vegetation coverage rate (VCR), soil depth, and leaf area index (*LAI*). The data were obtained from the National Glacial Tundra Desert Science Data Center (http://www.ncdc.ac.cn/portal/, accessed on 10 October 2022), the U.S. Geological Survey (https://ladsweb.modaps.eosdis.nasa.gov/search/, accessed on 10 October 2022), the Ecology Discipline Data Center of the Chinese Academy of Sciences (http://www.nesdc.org.cn/, accessed on 10 October 2022), and the wheatA small malt-agrometeorological big data system (v1.5.1) software (Wheat Bud Big Data Information (Ningbo) Co., LTD, Zhejiang, China) (It integrates multiple sections of the agricultural production, markets, meteorology, soil, marine and environmental monitoring) (http://www.wheata.cn/, accessed on 10 October 2022). The information of each variable is summarized in Table 1.

### 2.2. Methods

The overall empirical procedure is shown in Figure 2 below.

In this study, two models were used to make comparative predictions of the soil moisture at different depths. In Model A, the variables are mainly integrated into the water balance equation, and the depth coefficient γ is derived by the co-integration, and then the soil moisture prediction at different depths is performed. Model B directly uses the seasonal ARIMA model for the soil moisture prediction at different depths.

#### 2.2.1. A Soil Moisture Prediction Model, Based on the Depth and Water Balance Equation

The water variation in the grassland is cyclic, with the water vapor transported through precipitation, infiltration and evaporation. The water balance equation of the soil-vegetation-atmosphere system, without considering the anthropogenic factors is [37]:(1)$$\Delta \beta_{t}=\beta_{t}-\beta_{t-1}=P_{t}+G_{u,t}+R_{in,t}-\left(Et_{a,t}+G_{d,t}+R_{out,t}+IC_{store,t}\right)$$

In Equation (1), $\beta_{t}$ is the soil moisture in period *t* and Δβ is the variation of the soil moisture. At time *t*, $P_{t}$ is precipitation, $G_{u,t}$ and $G_{d,t}$ are the groundwater capillary rise and soil water infiltration. $Et_{a,t}$ is the soil evaporation, which is expressed by SE in this study. $R_{in,t}$ and $R_{out,t}$ are the inlet and outlet runoff, and $IC_{store,t}$ is the vegetation interception flow.

In typical grassland areas, the runoff and seepage are weak, and the precipitation and evapotranspiration represent the vast majority of the water balance [38]. The grassland of Xilingol League is located in the plain area, as the terrain is flat, the precipitation distribution is unbalanced, the intensity of the precipitation is generally low, the water circulation is mainly through a vertical exchange and the inlet and outlet runoff of the whole grassland can be regarded as roughly equal, so for the simplification of the model we assume that $R_{in,t}-R_{out,t}=0$. Since the data of the groundwater resource circulation are not available, the groundwater and soil contact, as well as the exchange water resources can also be assumed to be roughly equal, which means that $G_{u,t}-G_{d,t}=0$. Therefore, the amount of the soil moisture change is mainly related to the precipitation, evaporation and vegetation interception flow.

The vegetation interception flow is a variable that is mainly positively corelated with the precipitation, vegetation coverage rate and the leaf area index [39,40], which means that the higher the precipitation, the vegetation cover rate and the leaf area index, the higher the vegetation interception flow. It can be expressed as [41]:(2)$$IC_{store,t}=c_{p,t}\times IC_{max}\times \left(1-e^{\frac{-k\times P_{t}}{IC_{max}}}\right)$$

In Equation (2), $c_{p,t}$ denotes the vegetation coverage rate in period t, which is represented by VCR in this study; k is the correction factor for the vegetation density, which is related to the leaf area index; $P_{t}$ is the precipitation. $IC_{max}$ is the maximum interception of the particular vegetation, which can be estimated by the *LAI* [42], whose expression is:(3)$$IC_{max}=0.935+0.498\times LAI-0.00575\times LAI^{2}$$

Since *LAI*, P, *SE* and VCR are all time series, they are greatly influenced by the seasonal environment. Therefore, the seasonal ARIMA model is used for the prediction, and the specific steps are as follows:Determine the stationarity of the series. Testing the stationarity of the series with the unit root test. If the series is non-stationary, it needs to be differenced to eliminate the series fluctuations to make the data stationary, and extract the effective information in the series;Model order determination. The optimal order is firstly determined by the trailing and truncation of the autocorrelation function (ACF) and the partial autocorrelation function (PACF) (the ACF and PACF are two statistics that describe the degree of correlation between the current value of a time series and its past values, they are both used to describe the autocorrelation of the time series itself, the difference is that the PACF controls for some intermediate periods that are constant), and then the final parameters are determined by the trial-and-error method, based on the Akaike information criterion (AIC) in the approximate range of the initially determined parameters (the AIC is a measure of the goodness of fit of a statistical model, and the smaller the value of the AIC, the better the model);Model testing. The white noise test is for testing the residuals of the model. If there are insignificant parameters, they need to be removed and the model structure is readjusted. The white noise test ensures that the model can fully extract the relevant information of the series.

As a result, we obtain the prediction model as: $\beta_{t+1}=\beta_{t}+\Delta \beta$. Considering the effect of the depth h variation on the soil moisture, the depth parameter γ is introduced, and finally the soil moisture prediction model, based on the depth and water balance equation is obtained, as follows:(4)$$\beta_{h,t+1}=\alpha \beta_{h,t}+\gamma \times \Delta \beta =\beta_{h,t}+\gamma_{h}\times \left[P_{t}-\left(SE_{t}+IC_{store,t}\right)\right]$$

In Equation (4), α is the adjustment coefficient, $\beta_{h,t}$ is the soil moisture at depth *h* and time *t*, $\gamma_{h}$ is the depth parameter at depth *h*, $P_{t}$ is the precipitation, $SE_{t}$ is the soil evaporation, $IC_{store,t}$ is the vegetation interception flow.

#### 2.2.2. A Soil Moisture Prediction Model, Based on the Seasonal ARIMA Model

The soil moisture is a time series that only considers its own variation and changes with the months, which has certain seasonal characteristics. The univariate prediction of the soil moisture is thus validated for comparison using a seasonal ARIMA model.

In practice, the stationary process and parameter adjustment process for the time series is very time consuming. It is necessary to keep the data stationary when constructing the ARMA or ARIMA models, and the process of determining the *p* and q values is tedious and it is not easy to reach the exact values. The auto.arima function can eliminate the process of determining the *p* and q values and make the model more acceptable to the data [43]. Therefore, this study uses the auto.arima function to determine each parameter.

The seasonal ARIMA (p, d, q) ($P_{S}$, $D_{S}$, $Q_{S}$) model is based on the ARIMA (p, d, q) model and is used to deal with the series where there is a significant periodicity, due to the seasonal variations. The seasonal ARIMA model automatically determines the d, *p* and q values through the auto.arima function and fits the model on a univariate series, from which the soil moisture inputs at different depths can be predicted.

#### 2.2.3. Error Estimation

For the purpose of assessing the validity and accuracy of the model, this study uses the *MSE* (mean squared error), the *RMSE* (root mean squared error), the *MAE* (mean absolute error) and the *MAPE* (mean absolute percentage error) to evaluate the model.
(5)$$MSE=\frac{1}{n}\overset{n}{\underset{i=1}{\sum }}{\left(\hat{y_{i}}-y_{i}\right)}^{2}$$
(6)$$RMSE=\sqrt{\frac{1}{n}\overset{n}{\underset{i=1}{\sum }}{\left(\hat{y_{i}}-y_{i}\right)}^{2}}$$
(7)$$MAE=\frac{1}{n}\overset{n}{\underset{i=1}{\sum }}\left|\hat{y_{i}}-y_{i}\right|$$
(8)$$MAPE=\frac{100}{n}\overset{n}{\underset{i=1}{\sum }}\left|\frac{\hat{y_{i}}-y_{i}}{y_{i}}\right|$$

In the above equation *n* is the total number of test sets, $\hat{y_{i}}$ is the predicted value and $y_{i}$ is the actual value.

## 3. Results and Discussion

### 3.1. Model Construction and Prediction

#### 3.1.1. The Soil Moisture Prediction Model, Based on the Depth and Water Balance Equation

To begin with, this study conducted a descriptive statistical analysis of the variables involved, as shown in Table 2. The main independent variables involved were the soil evaporation (SE), precipitation (P) and normalized difference vegetation index (NDVI). Among them, the coefficient of variation of the P is the largest, with a range of 891.79, which means that there is a large amount of precipitation in the rainy season and no precipitation for a month during the dry season, and the precipitation fluctuates. The coefficient of the variation of the *NDVI* is the smallest, and the minimum value −0.02 means that the ground cover is highly reflective of visible light, such as water and snow. The coefficient of variation of the SE is in between that of the P and the *NDVI*.

The dependent variable to be studied is the soil moisture at different depths, and the descriptive statistics table of known data is shown below in Table 3. The coefficient of the variation of soil moisture at different depths has been clearly calculated from the table: 100 cm > 10 cm > 40 cm > 200 cm. The external factors, such as the vegetation, can have a significant effect on the soil moisture [44], especially near the surface at a 10 cm depth, hence the large variation in the soil moisture at 10 cm. As the surface deepens (0–50 cm), the spatial heterogeneity of the soil moisture decreases [45], so that the variation of the soil moisture at 40 cm decreases. Until 100 cm, the soil is more active [46], and the soil moisture variation is at its maximum. As the soil depth increases, the stability of the soil moisture continues to increase [47], so that the 200 cm soil moisture varies the least.

In order to visualize the variation of the soil moisture at different depths, a line graph was drawn, as shown in Figure 3. It is also clear from this graph that the soil moisture at 200 cm is almost a straight line with small variations. The soil moisture at 10 cm fluctuates back and forth within a small interval. The variation in the soil moisture was greater at 40 cm and 100 cm and was similar for both.

According to the soil moisture prediction model, based on the depth and water balance equation constructed above, the data of the VCR are needed to be obtained. However, the VCR has only partial data from August 2020 to 2022, and the data are relatively few. There is a close relationship between the *NDVI* and VCR [48], so this study uses a regression model to construct a model of the *NDVI* and VCR. Then, the VCR from 2012 to July 2020 was obtained by substituting the known *NDVI* from 2012 to July 2020 into the regression equation.

In order to investigate which model is more suitable for fitting the *NDVI* and VCR, linear regression, quadratic regression and cubic regression models were constructed, and the model summary is listed in Table 4. It shows that the F-test of the linear, quadratic and cubic regressions are significant. Since the adjusted R-squared of the cubic regression is the largest and all of the coefficients in the cubic regression are significant at the 0.05 level, the method used in this paper is the cubic regression.

The regression equation for the *NDVI* and VCR can be obtained from the parameter estimates in Table 4 as:(9)$$\hat{\mathrm{VCR}}=0.008-0.894NDVI+5.827NDVI^{2}-5.778NDVI^{3}$$

Upon completing the data on the VCR, the line graphs of the *SE*, P, *LAI* and VCR were plotted in Figure 4, from which it was clear that there was a direct and significant seasonal trend in these four variables. From this, the seasonal ARIMA model can be used to estimate the prediction of the time series.

The predicted data of the *SE*, P, *LAI* and VCR by the seasonal ARIMA model are shown in Table 5. From the predicted data and model calculations, $IC_{max}=1.419$. To simplify the model, we assume that the vegetation density has little effect, so the value of k is set to 1 by default in this study. $IC_{store}$ and Δβ can be calculated by substituting the data, both of which are also shown in Table 5. 

Since $IC_{store}$ is directly related to the P and VCR, $IC_{store}$ is 0 when there is no rainfall in the dry season and no vegetation covered by snow and ice in winter. According to the model constructed above, the soil moisture $\beta_{h,\;t+1}$ at depth *h* of the nth + 1st month can be calculated by the soil moisture variation Δβ, the depth coefficient γ and the soil moisture $\beta_{h,t}$ of the nth month. The regression models were performed for Δβ and the soil moisture at different depths, from which coefficients γ were obtained for the different depths, and the *p*-values of the tests for the coefficients were all less than 0.05, indicating that all coefficients were significant. The resulting soil moisture prediction data are shown in Table 6. 

#### 3.1.2. The Soil Moisture Prediction Model, Based on the Seasonal ARIMA Model

In order to compare the accuracy of the predictions, a soil moisture prediction model, based on the seasonal ARIMA model, was constructed in this study. The auto.arima function is used to select the model with the smallest AIC and BIC, and the constructed 10 cm, 40 cm, 100 cm and 200 cm models are shown in Equations (10)–(13).
(10)$$\nabla_{12}\hat{\beta_{10,t}}=\frac{\left(1-0.7593B^{12}\right)}{\left(1-0.5852B\right)}\varepsilon_{t}$$
(11)$$\nabla \hat{\beta_{40,t}}=\frac{\left(1-0.9869B\right)}{\left(1-1.0677B+0.2250B^{2}\right)\left(1-0.4081B^{12}-0.2313B^{24}\right)}\varepsilon_{t}$$
(12)$$\nabla \hat{\beta_{100,t}}=\left(1+0.5308B\right)\varepsilon_{t}$$
(13)$$\nabla^{2}\hat{\beta_{200,t}}=\frac{\left(1-0.9237B-0.6746B^{2}+0.8232B^{3}\right)\left(1-0.3649B^{12}+0.4964B^{24}\right)}{\left(1-0.7436B-0.59230B^{2}+0.8941B^{3}\right)\left(1-0.6271B^{12}\right)}\varepsilon_{t}$$

In Equations (10)–(13), β is the soil moisture at the different depths, *B* is the lag operator, $\varepsilon_{t}$ is the error term, and ∇ means the difference.

Meanwhile, the QQ plots of the fitted models with the residuals are plotted separately, as shown in Figure 5.

Most of the residuals in Figure 5 closely follow the median trend line. The *p*-values of the Ljung–Box test are greater than 0.05, none of which rejects the original hypothesis (the Ljung–Box test is used to test the randomness of the time series). These indicate that the residuals meet the normality assumption and pass the white noise test, then the model can be considered to have adequately fitted the data and can be used in the next step of the forecasting process. The final predicted data were obtained, as shown in Table 7.

The soil moisture predictions, based on the seasonal ARIMA model, were obtained by combining the predicted data with the original data, while 95% confidence intervals were plotted for the predicted data, as shown in Figure 6.

From Figure 6, it can be seen that the predicted values and the confidence intervals of the soil moisture at a 10 cm depth and 40 cm depth showed periodic fluctuations, but the confidence intervals were larger at the 40 cm depth. At 100 cm, the model predicted a horizontal straight line with a significant increase in uncertainty and a flared spread of the confidence intervals. By the 200 cm depth, the variation in the soil moisture was smaller and the confidence intervals were not significant.

### 3.2. Model Comparison

In order to see the trend of the prediction results more clearly, a comparison chart of the prediction trend is drawn, as shown in Figure 7. Model A represents the soil moisture prediction model, based on the depth and water balance equation, and Model B represents the soil moisture prediction model, based on the seasonal ARIMA model.

The predicted data and the known data were combined to draw a line graph, as shown in Figure 7. It can be seen from Figure 7 that the prediction results of Model A show a fluctuating decreasing trend at 10 cm, 40 cm and 200 cm, and at 100 cm, the prediction value increases with the previous trend, while Model B shows a different change. At a depth of 10 cm, Model A predicts smaller periodic fluctuations in the soil moisture with a small range of variation, while Model B predicts a larger trend of the periodic fluctuations in the soil moisture with a large range of variation. At a depth of 40 cm, Model A predicts a fluctuating decline in the soil moisture with time, but with large fluctuations, while Model B predicts that the soil moisture fluctuates back and forth within an interval with small changes. At a depth of 100 cm, Model A predicts that the soil moisture will continue to fluctuate upward with the previous trend, while Model B predicts that the soil moisture will remain constant. At a depth of 200 cm, Model A predicts a small fluctuating decrease in the soil moisture, while Model B predicts a much larger fluctuating decrease in the soil moisture. On the whole, the soil moisture showed a regular pattern of periodic changes involving large fluctuations, which is consistent with Cai’s findings [49]. Model A showed a more moderate fluctuation at 10 cm and 200 cm, rising at 100 cm, according to the previous trend, and fluctuating downward at other depths. Model B showed a clear cyclical pattern at 10 cm and 40 cm, it predicted to remain constant at 100 cm, and declined more at 200 cm.

Using the data from January 2012 to December 2020 as the training set, and January 2021 to March 2022 as the test set, the prediction errors of Model A and Model B were obtained, as shown in Table 8.

It is clear from Table 8 that, as a whole, the prediction errors of both models increase and then decrease with the increasing depth, with the largest prediction error at 40 cm and the smallest prediction error at 200 cm. This may be due to the regularity of other influencing factors close to the surface, which creates a strong cycle. As the depth increases, the soil moisture is less affected by the external influences until 200 cm, when it basically maintains a stable state, so the prediction error at 200 cm is the smallest.

The error of Model A is larger than Model B at 10 cm because the soil at 10 cm is affected by the atmosphere, vegetation and other factors [50,51], creating stronger cyclical fluctuations, thus it is more consistent with the time series forecasting methods. However, with the increase of the depth, the influence of uncertainties becomes larger and the advantage of Model A comes into play. The errors of Model A were smaller than Model B at 40 cm, 100 cm and 200 cm depths, and Model A has a high accuracy in predicting the soil moisture with a weaker periodicity and regularity.

The prediction residuals for the different depths are plotted as area plots, as shown in Figure 8. From Figure 8, it can be seen that the error fluctuation of Model A at 10 cm, is larger than that of Model B, resulting in a lower overall prediction accuracy at 10 cm, than that of Model B. However, the prediction residuals of Model A at 40 cm, 100 cm and 200 cm, are smaller than that of Model B, with smaller prediction errors and a higher prediction accuracy.

The prediction of Model B was better at 10 cm; the prediction of Model A was better at 40 cm, 100 cm and 200 cm. This demonstrates the effectiveness of the soil moisture prediction model, based on the depth and water balance equation, which exhibits a satisfactory prediction performance and prediction accuracy. More importantly, in the soil moisture prediction, the more accurate prediction values can be obtained using the different methods. In this study, for example, a soil moisture prediction model, based on the seasonal ARIMA model, can be used at 10 cm, and a soil moisture prediction model, based on the depth and water balance equation can be used at 40 cm, 100 cm and 200 cm.

## 4. Conclusions

In summary, this study proposes a soil moisture prediction model, based on the depth and water balance equation that combines the water balance equation and the seasonal ARIMA model. Compared to predicting the soil moisture using methods, such as machine learning [52,53,54], the model can explain the variables and the model principles with a theoretical significance. Compared to the monitoring and prediction of the soil moisture using remote sensing data [55,56,57], the model is not only able to predict the soil moisture at the surface, but also at different depths, based on the inclusion of the depth coefficients. The experimental results showed that the model proposed in this study was able to provide a higher prediction accuracy for the soil moisture prediction at 40 cm, 100 cm and 200 cm, compared to the soil moisture prediction model, based on the seasonal ARIMA model. The RMSEs of the model were 2.853, 5.666, 4.732 and 0.682 at 10 cm, 40 cm, 100 cm and 200 cm, respectively, and the overall predicted values were close to the true values, which could accurately predict the soil moisture.

### 4.1. Theoretical Contributions

The soil moisture plays a key role in the sustainability of the grasslands, as predicting the soil moisture helps to improve the drought prediction and agricultural crop production, leading to the appropriate irrigation, grazing plans and water management. The physics-based process model is more complex, and the absence or lack of information can introduce some errors into the soil moisture prediction. Since the equations are built on the assumption of the independence of local laws, they are not suitable for the medium to long term predictions. The data-driven empirical models, such as machine learning, are questionable because the principles of prediction are not available, and a small amount of data can lead to underfitting and other problems. To address these shortcomings, this study proposes a novel hybrid model for predicting the soil moisture, a soil moisture prediction model, based on the depth and water balance equation, which integrates the water balance equation with the seasonal ARIMA model and introduces a depth parameter to predict the soil moisture at different depths. The model has a high prediction accuracy at 40 cm, 100 cm and 200 cm. This study also proposes that different models can be used for the prediction at different depths, for example, using the soil moisture prediction model, based on the seasonal ARIMA model at 10 cm and the soil moisture prediction model, based on the depth and water balance equation at 40 cm, 100 cm and 200 cm, from which more accurate prediction values can be obtained.

### 4.2. Policy Implications

At present, due to population growth and economic development, the intensity of land use has been increasing, leading to the destruction of the ecological environment of the land resources and the emergence of more and more fragile ecological areas. This seriously threatens the ecological security of the regional land resources and directly affects the sustainable development of the region. The scientific evaluation of the ecological safety of the regional land resources and the rational use of land are related to the fate of the country and social development. The soil moisture plays an important role in the research and application of the climate, environment and ecology, which can well reflect the ecological environment quality status of the region, and is an important indicator for monitoring the surface environment. The accurate prediction of the soil moisture can provide an important theoretical basis for sustainable land use, water resources planning and management and agricultural production. This has important implications for the regional ecosystem conservation.

Different types of soils require different policies to protect the regional ecosystems. In this study, the Xilingol League grassland is used as an example. The prediction shows that by December 2023, the soil moisture at 10 cm will show a strong periodic fluctuation as before, the soil moisture at 40 cm and 200 cm will fluctuate down, and the soil moisture at 100 cm will maintain the previous trend of rising. Overall, the moisture at 40 cm, 100 cm and 200 cm, is roughly 3.18, 7.79 and 11.72 times that of the moisture at 10 cm. The Xilingol League grassland in Inner Mongolia has a low precipitation and uneven distribution, and in the process of promoting its sustainable development, the ecological protection should be the center, and some local areas with severely degraded grasslands should be the priority areas for construction and protection work. In order to prevent the increase of the soil aridity and thus prevent the land desertification and soil slabbing, the appropriate irrigation plans can be made and appropriate grazing can be achieved to realize the scientific management of the grasslands [58]. This promotes the ecological stability and the sustainable development of the grasslands, while rationalizing land use.

### 4.3. Limitations

Soil moisture is a complex nonlinear coupled system, which is greatly influenced by the external environment and other factors. The limitations of this study mainly exist in the following two points. First, the nature of the different soils is different. This study was carried out only for the kastanozems in the grasslands of Xilingol League, and there is a lack of research on the soil moisture relationships in other types of soils. Second, in order to simplify the study, more assumptions were set in this study, which may be different from the real situation.

### 4.4. Future Research Perspectives

In view of the shortcomings of the research in this paper, the paper can be improved in the following two aspects when the conditions allow. First, the different soils are classified, the exclusive models are constructed for the different types of soils and different depths, and professional policies are provided, according to the regional conditions. Second, the soil moisture is influenced by more external factors, and since the parameters are related to the fixed-point data, the association between the local factors can be studied in depth, and the association between the soil moisture and its influencing factors in the different regions can also be explored. Thus, the constraint assumptions can be released, as much as possible, to improve the generalization ability of the model and form a model that can be further promoted. Subsequent research can be conducted, based on the above two aspects, to obtain more accurate soil moisture prediction values, which can then promote sustainable land use and the sustainable development of the regional ecology.

## Figures and Tables

**Figure 1 ijerph-20-01374-f001:**
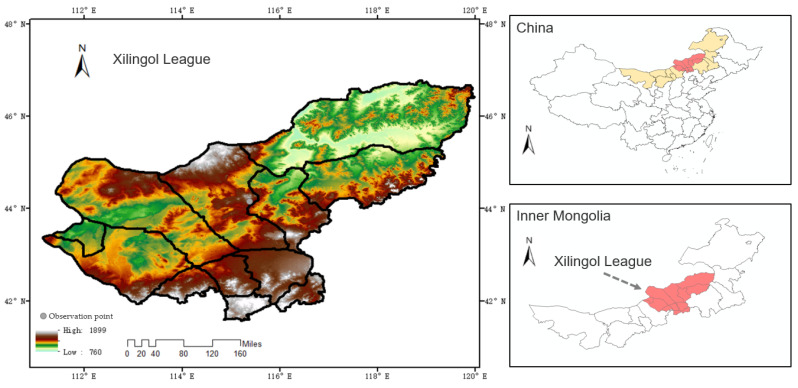
Study area of the Xilingol League grassland.

**Figure 2 ijerph-20-01374-f002:**
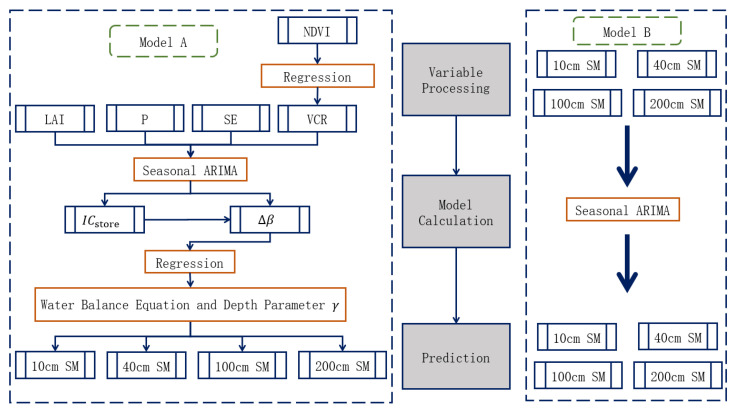
Empirical flow chart of the soil moisture prediction model.

**Figure 3 ijerph-20-01374-f003:**
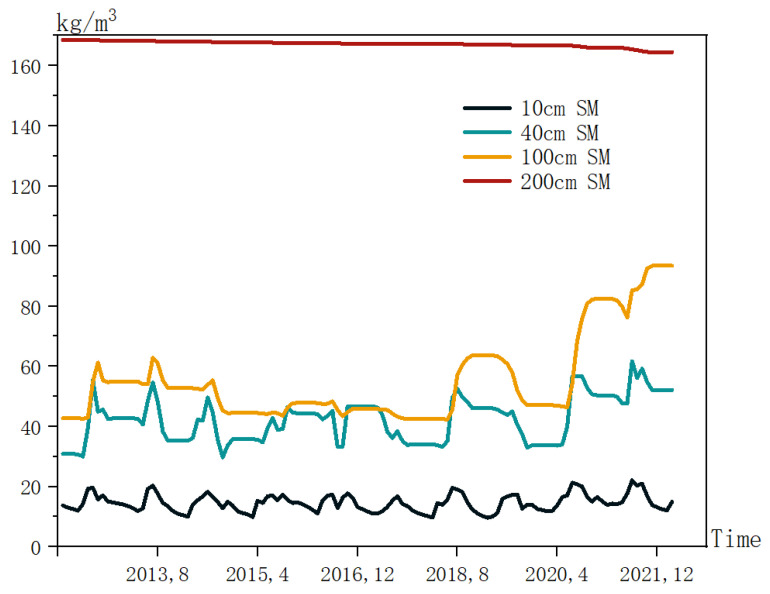
Line graph of the historical soil moisture data at different depths.

**Figure 4 ijerph-20-01374-f004:**
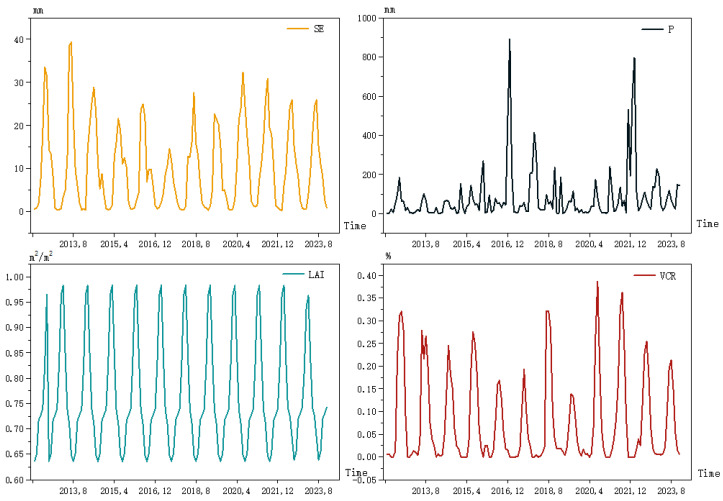
Historical data of the SE, P and LAI and the fitted VCR data.

**Figure 5 ijerph-20-01374-f005:**
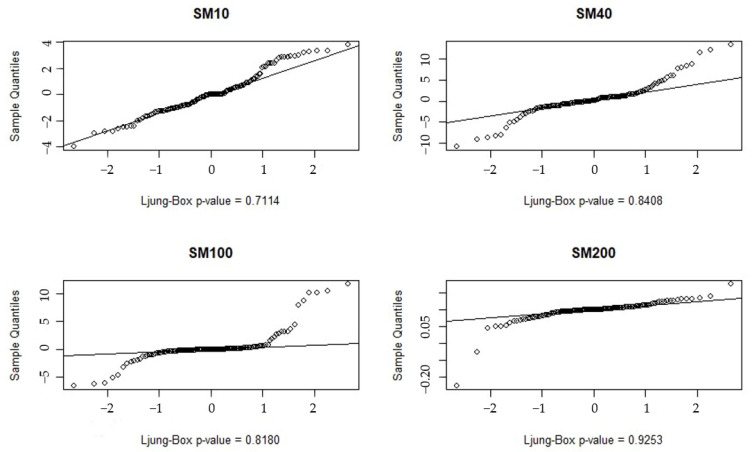
QQ plots of the predicted residuals of the seasonal ARIMA model at the different depths.

**Figure 6 ijerph-20-01374-f006:**
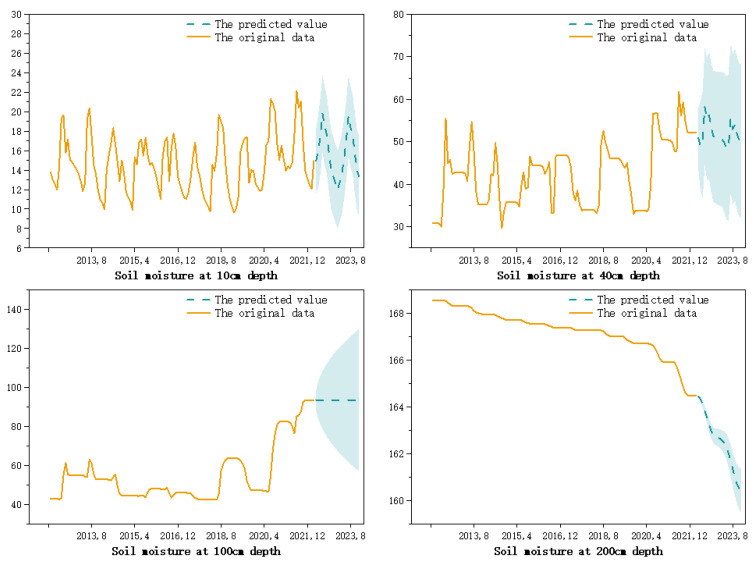
Soil moisture prediction results, based on the seasonal ARIMA model with a 95% confidence interval.

**Figure 7 ijerph-20-01374-f007:**
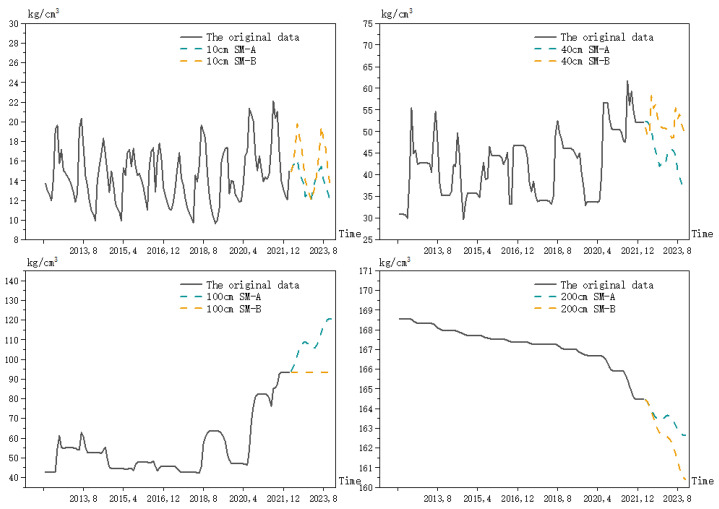
Comparison of the overall trend of the soil moisture prediction by the different models.

**Figure 8 ijerph-20-01374-f008:**
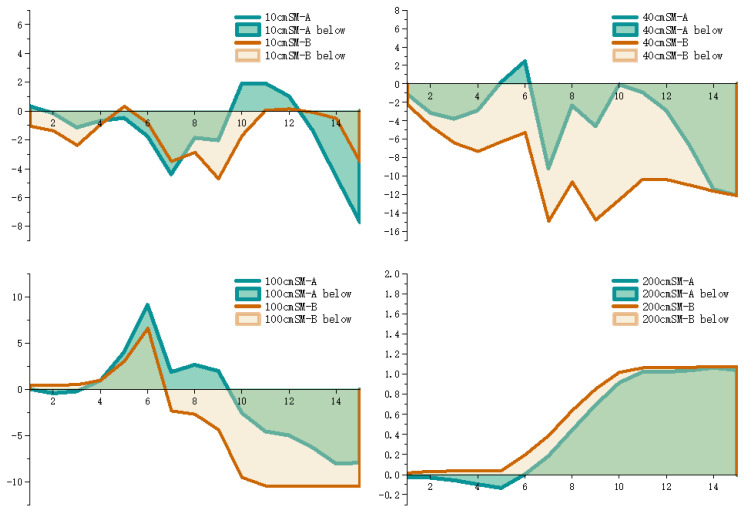
Predicted residual area charts of different models at different depths.

**Table 1 ijerph-20-01374-t001:** Observed values of each variable.

Variables	Longitude	Latitude	Starting Time	End Time	Unit	N
LAI	115.375	44.125	2012.1	2022.3	m^2^/m^2^	123
NDVI					—	123
SE					mm	123
P					mm	123
10 cm SM					kg/m^3^	123
40 cm SM					kg/m^3^	123
100 cm SM					kg/m^3^	123
200 cm SM					kg/m^3^	123
VCR			2020.7		%	21

**Table 2 ijerph-20-01374-t002:** Descriptive statistics of the P, NDVI and SE, from January 2012 to March 2022.

Independent Variable	Unit	Range	Minimum	Maximum	Mean	Standard Deviation	Coefficient of Variation
P	mm/month	891.79	0	891.79	80.83	143.07	1.77
NDVI	—	0.62	−0.02	0.61	0.24	0.15	0.63
SE	mm/month	38.99	0.27	39.26	9.87	9.75	0.99

**Table 3 ijerph-20-01374-t003:** Descriptive statistics of the soil moisture data at different depths, from January 2012 to March 2022.

Dependent Variable	Unit	Minimum	Maximum	Mean	Standard Deviation	Coefficient of Variation
10 cm SM	kg/m^3^	9.64	22.1	14.57	2.85	0.20
40 cm SM	kg/m^3^	29.71	61.7	42.38	7.48	0.18
100 cm SM	kg/m^3^	42.36	93.45	55.35	14.54	0.26
200 cm SM	kg/m^3^	164.48	168.56	167.22	0.98	0.01

**Table 4 ijerph-20-01374-t004:** Analysis of the different regression fitting results.

Model	Models Summary		Parameter Estimates			
	Adjust R-Squared	Prob (F-Test)	c	b1	b2	b3
linear	0.764	0.000	−0.055 *	0.607 ***		
quadratic	0.826	0.000	−0.011	0.067	0.941 **	
cubic	0.864	0.000	0.008	−0.894 *	5.827 **	−5.778 *

* Note: “*” means significant at 0.05; “**” means significant at 0.01; “***” means significant at 0.001.

**Table 5 ijerph-20-01374-t005:** Prediction data for SE, P, LAI and VCR, and the fitted values of *IC_store_* and Δ*β*.

Year	Month	SE	P	LAI	VCR	*IC_store_*	Δ*β*
2022	4	10.8614	0.0000	0.7270	0.0140	0.0000	−10.8614
	5	18.3533	65.4249	0.7390	0.0319	0.0452	−16.2177
	6	23.1401	77.9889	0.8270	0.0919	0.1304	−20.6709
	7	27.0137	89.3870	0.9640	0.2348	0.3332	−24.3674
	8	17.4328	69.7163	0.9830	0.2893	0.4106	−15.5195
	9	13.1105	81.2433	0.8470	0.1724	0.2447	−10.6471
	10	8.8015	64.5582	0.7380	0.0582	0.0825	−6.7321
	11	2.3674	230.9857	0.7100	0.0102	0.0144	5.3177
	12	0.9876	125.7876	0.6480	0.0007	0.0010	3.2044
2023	1	0.6344	257.7069	0.6364	0.0059	0.0083	7.9474
	2	0.7005	292.5912	0.6507	0.0021	0.0030	9.0495
	3	4.5560	88.0504	0.7168	0.0120	0.0170	−1.6379
	4	9.9031	57.6033	0.7270	0.0186	0.0264	−8.0093
	5	18.0552	83.1367	0.7390	0.0420	0.0596	−15.3435
	6	23.0474	83.1367	0.8270	0.1080	0.1532	−20.4294
	7	26.9849	83.1367	0.9640	0.2814	0.3992	−24.6129
	8	17.4238	83.1367	0.9830	0.3535	0.5016	−15.1542
	9	13.1078	83.1367	0.8470	0.2137	0.3033	−10.6398
	10	8.8006	83.1367	0.7380	0.0701	0.0994	−6.1288
	11	2.3671	83.1367	0.7100	0.0078	0.0111	0.3930
	12	0.9875	83.1367	0.6480	0.0002	0.0002	1.7835

**Table 6 ijerph-20-01374-t006:** Predicted values of the soil moisture, based on the depth and water balance equation.

Year	Month	10 cm SM	40 cm SM	100 cm SM	200 cm SM
		*γ*_10_ = −0.1069	*γ*_40_ = −0.1357	*γ*_100_ = −0.0652	*γ*_200_ = −0.0018
2022	4	15.12	52.35	94.11	164.44
	5	15.59	52.23	96.09	164.32
	6	15.91	51.57	98.81	164.15
	7	16.12	50.45	102.15	163.94
	8	14.94	47.35	105.22	163.71
	9	14.28	45.48	107.23	163.56
	10	13.76	44.12	108.58	163.47
	11	12.41	41.95	108.80	163.42
	12	12.68	42.66	108.14	163.47
2023	1	12.21	42.26	107.35	163.51
	2	12.16	42.73	106.09	163.59
	3	13.39	44.89	105.43	163.66
	4	14.05	45.63	106.09	163.63
	5	14.76	46.00	107.77	163.54
	6	15.17	45.49	110.40	163.38
	7	15.42	44.46	113.73	163.17
	8	14.18	41.26	116.81	162.94
	9	13.56	39.46	118.79	162.79
	10	12.98	38.02	120.09	162.70
	11	12.22	36.66	120.58	162.65
	12	12.08	36.50	120.43	162.65

**Table 7 ijerph-20-01374-t007:** Predicted values of the soil moisture, based on the seasonal ARIMA model.

Year	Month	10 cm SM	40 cm SM	100 cm SM	200 cm SM
2022	4	14.92	50.85	93.41	164.43
	5	15.94	49.00	93.41	164.34
	6	17.48	49.40	93.41	164.11
	7	19.81	58.23	93.41	163.85
	8	18.51	55.37	93.41	163.54
	9	17.64	56.20	93.41	163.23
	10	15.71	53.16	93.41	162.94
	11	14.00	51.28	93.41	162.78
	12	13.34	50.99	93.41	162.72
2023	1	12.60	50.81	93.41	162.69
	2	11.98	50.68	93.41	162.64
	3	12.73	50.57	93.41	162.57
	4	13.62	49.84	93.41	162.45
	5	15.18	48.55	93.41	162.32
	6	17.03	48.63	93.41	162.02
	7	19.54	55.45	93.41	161.69
	8	18.36	52.96	93.41	161.32
	9	17.55	54.00	93.41	160.97
	10	15.66	51.72	93.41	160.66
	11	13.97	50.32	93.41	160.50
	12	13.32	50.18	93.41	160.41

**Table 8 ijerph-20-01374-t008:** Comparison of the soil moisture prediction errors of the different models.

	10 cm SM		40 cm SM		100 cm SM		200 cm SM	
	A	B	A	B	A	B	A	B
RMSE	2.853	**2.138**	**5.666**	10.072	**4.732**	6.981	**0.682**	0.739
MSE	8.138	**4.571**	**32.108**	101.437	**22.392**	48.729	**0.465**	0.547
MAE	2.083	**1.603**	**4.253**	9.379	**3.717**	5.584	**0.518**	0.578
MAPE	13.369	**9.299**	**7.969**	17.529	**4.231**	6.207	**0.315**	0.351

Note: The values marked in **bold** are the smaller error values at the same depth.

## Data Availability

Data sources are contained within the article.

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
