# Peer review of "A Soil Moisture Prediction Model, Based on Depth and Water Balance Equation: A Case Study of the Xilingol League Grassland"

_ijerph, 2023, doi:10.3390/ijerph20021374_

Round 1

Reviewer 1 Report

This is a very intersting research topic, the author used different model to predict soil moisture. As the author says, accurate prediction of soil moisture can contribute to the scientific management of grasslands. However, there are major defects in the method of this study. The reliability verification of the model is obviously insufficient.

Author Response

We thank the reviewer for your suggestions on this study. We have added regression models to derive depth coefficients and added article citations to make the manuscript more reliable. We hope that our revised manuscript will satisfy you!

Reviewer 2 Report

L 51-57: Please cite the statements

L 76 – 80: Please reword and cite the statement

Section 2.1 – Please provide a monthly weather chart/diagram with temperature, precipitation,  etc

L 148-155: These assumptions are very farfetched, please cite them appropriately or provide data taken by authors

L 160-161: Higher vegetation and LAI with low precipitation seem very unreasonable. Please provide proper sources to justify them. Also please justify using other parameters such as NDVI.

L 217: I think the standard deviation of NDVI (.15) is high considering NDVI ranges from +1 to -1

L 218: This explanation/conclusion is off. Please cite or change it.

L 224-232: These explanations are very farfetched, please cite them appropriately

L 341-343: Have the authors tried to determine the factors influencing at 40 cm depth and include them in the model to have a more robust model?

Author Response

First of all, we would like to thank you for your contribution to this study, and we have cited many articles to make the article more reliable in response to the issues you raised and the modified parts are highlighted in green. The following are the specific revisions in response to your comments.

Q1: L 51-57: Please cite the statements

A1: Thanks for the reviewer's suggestion. In this regard we have cited the following articles to make the manuscript more reliable. We have marked the answer to this question in line 51.

(1) Li, P.; Zha, Y.; Shi, L.; Tso, C.-H. M.; Zhang, Y.; Zeng, W. Comparison of the Use of a Physical-Based Model with Data Assimilation and Machine Learning Methods for Simulating Soil Water Dynamics. Journal of Hydrology 2020, 584, 124692. https://doi.org/10.1016/j.jhydrol.2020.124692.

(2) Freeze, R. A.; Harlan, R. L. Blueprint for a Physically-Based, Digitally-Simulated Hydrologic Response Model. Journal of Hydrology 1969, 9 (3), 237–258. https://doi.org/10.1016/0022-1694(69)90020-1.

(3) Elshorbagy, A.; Corzo, G.; Srinivasulu, S.; Solomatine, D. P. Experimental Investigation of the Predictive Capabilities of Data Driven Modeling Techniques in Hydrology - Part 1: Concepts and Methodology. Hydrol. Earth Syst. Sci. 2010, 14 (10), 1931–1941. https://doi.org/10.5194/hess-14-1931-2010.

(4) Dai, J.-Y.; Cheng, S.-T. Modeling Shallow Soil Moisture Dynamics in Mountainous Landslide Active Regions. Front. Environ. Sci. 2022, 10, 913059. https://doi.org/10.3389/fenvs.2022.913059.

Q2: L 76 – 80: Please reword and cite the statement

A2: Thanks for the reviewer's suggestion. For this we have cited the following articles and rewritten the manuscript. We have marked the answer to this question in line 75.

(1) Zhu, Q.; Luo, Y.; Zhou, D.; Xu, Y.-P.; Wang, G.; Tian, Y. Drought Prediction Using in Situ and Remote Sensing Products with SVM over the Xiang River Basin, China. Nat Hazards 2021, 105 (2), 2161–2185. https://doi.org/10.1007/s11069-020-04394-x.

(2) Hosseini, R.; Newlands, N.; Dean, C.; Takemura, A. Statistical Modeling of Soil Moisture, Integrating Satellite Remote-Sensing (SAR) and Ground-Based Data. Remote Sensing 2015, 7 (3), 2752–2780. https://doi.org/10.3390/rs70302752.

Q3: Section 2.1 – Please provide a monthly weather chart/diagram with temperature, precipitation, etc.

A3: Thanks for the reviewer's suggestion. The independent variables used in the article, such as monthly precipitation, are plotted in Figure 4 due to the need for the analysis, so they are not repeated. We have marked the answer to this question in line 267.

Q4: L 148-155: These assumptions are very farfetched, please cite them appropriately or provide data taken by authors.

A4: Thanks for the reviewer's suggestion. For this we have cited the following article and rewritten the manuscript. We have marked the answer to this question in line 139.

1.Chen Zuozhong. Typical grassland ecosystems in China. Science Press; Beijing, 2000.

Q5: L 160-161: Higher vegetation and LAI with low precipitation seem very unreasonable. Please provide proper sources to justify them. Also please justify using other parameters such as NDVI.

A5: We thank the reviewer for pointing out our errors, which we have corrected and cited the sources of the individual equations to justify the other parameters. For example, NDVI and VCR are highly correlated, so NDVI was used to fit VCR, as can be confirmed by the literature [44]. We have marked the answer to this question in line 150.

Q6: L 217: I think the standard deviation of NDVI (.15) is high considering NDVI ranges from +1 to -1.

A6: Thanks for the reviewer's suggestion. We verified this data and the high standard deviation of NDVI may be due to the particular geography. For this we further calculated the coefficient of variation to measure the fluctuations between the respective variables. We have marked the answer to this question in line 222.

Q7: L 218: This explanation/conclusion is off. Please cite or change it.

A7: We thank the reviewer for pointing out our error, which we have revised and used the coefficient of variation to compare the magnitude of fluctuations between the three independent variables. We have marked the answer to this question in line 226.

Q8: L 224-232: These explanations are very farfetched, please cite them appropriately.

A8: Thanks for the reviewer's suggestion. For this we have cited the following articles and rewritten the manuscript. We have marked the answer to this question in line 230.

(1) Zhu, Q.; Lin, H. Influences of Soil, Terrain, and Crop Growth on Soil Moisture Variation from Transect to Farm Scales. Geoderma 2011, 163 (1–2), 45–54. https://doi.org/10.1016/j.geoderma.2011.03.015.

(2) Bogena, H. R.; Herbst, M.; Huisman, J. A.; Rosenbaum, U.; Weuthen, A.; Vereecken, H. Potential of Wireless Sensor Networks for Measuring Soil Water Content Variability. Vadose Zone Journal 2010, 9 (4), 1002–1013. https://doi.org/10.2136/vzj2009.0173.

(3) Suo, L.; Huang, M.; Zhang, Y.; Duan, L.; Shan, Y. Soil Moisture Dynamics and Dominant Controls at Different Spatial Scales over Semiarid and Semi-Humid Areas. Journal of Hydrology 2018, 562, 635–647. https://doi.org/10.1016/j.jhydrol.2018.05.036.

(4) Gao, L.; Shao, M. Temporal Stability of Soil Water Storage in Diverse Soil Layers. CATENA 2012, 95, 24–32. https://doi.org/10.1016/j.catena.2012.02.020.

Q9: L 341-343: Have the authors tried to determine the factors influencing at 40 cm depth and include them in the model to have a more robust model?

A9: Thanks for the reviewer's suggestion. Since we were predicting soil moisture at multiple depths, we did not consider the factor of soil moisture at 40 cm depth alone. We include this in our vision for future research to build proprietary models for different depths. We have marked the answer to this question in line 442.

Finally, thank you again for your contribution to this study.

Reviewer 3 Report

The paper entitled: A Soil Moisture Prediction Model Based on Depth and Water 2 Balance Equation: A Case Study of Xilingol League Grassland attempts to predict soil moisture at different depths using a combination of ARIMA model and a depth parameters. There are many not so major observations but which cumulate into a "major revision". The authors introduce this "depth parameter" which plays a central role in the model, but which is not explained. How was it computed? it is not clear if the authors used more than one soil profile for their study. The study area should be presented into a figure, the soil which was sampled should be described in terms of structure, texture, bedrock lithology etc. Many other observations can be found in the annotated manuscript. 

Author Response

Thank you for pointing out our errors. We have made corrections based on your annotated manuscript and the modified parts are highlighted in yellow. Since it is not possible to make changes on the PDF, here we have selected some important questions to reply. In response to your questions, we provide the following answers.

Q1: What is the depth parameter? How do you compute it. Please explain

A1: The depth parameter is the coefficient of variation at different depths. We used the test method before, now we add the regression model to make the results more accurate. The depth parameters were calculated by regression model, and the p-values of the tests for the fitted values of all parameters were less than 0.05 and were significant. We have marked the answers to the questions in line 278.

Q2: Some data sources' websites are not working.

A2: Data were obtained from the "China Optics Valley-Huawei Cup" the 19th China post-graduate mathematical contest in modeling (https://cpipc.acge.org.cn/cw/detail/4/2c90801583651d4101838850aace3087). The title description states that these data are monitored and provided by specialized agencies, but there is not much description of soil characteristics, such as how soil evaporation and soil humidity were measured etc. We have marked the answers to the questions in line 124.

Q3: Please add a figure in this section showing the geographical position of the study area. Is the formula (2) yours? If not, please add a reference.

A3: The study area has been mapped as shown in Figure 1. In addition, we have added citations to increase the reliability of the article, for example, the equations are cited to the sources. We have marked the answers to the questions in line 120.

Q4: isn't this value too low? Please check. I found a value around 14℃.

A4: The average annual temperature of Xilingol grassland is 0.96℃(https://baike.baidu.com/item/%E9%94%A1%E6%9E%97%E9%83%AD%E5%8B%92%E8%8D%89%E5%8E%9F/576299?fromModule=lemma_search-box). We have marked the answers to the questions in line 105.

Q5: please explain the abbreviation. ACF, PACF, AIC and Ljung-Box test

A5: ACF and PACF are two statistics that describe the degree of correlation between the current value of a time series and its past values, they are both used to describe the autocorrelation of the time series itself, the difference is that PACF controls for some intermediate periods that are constant. AIC is a measure of the goodness of fit of a statistical model, and the smaller the value of AIC, the better the model. We have marked the answers to the questions in line 166.

Ljung-Box test is to test the randomness of the time series. We have marked the answers to the questions in line 294.

These are terms used in time series.

Q6: please explain the notations.

A6: In Equations (10)-(14), B is the lag operator,  is the error term, and ∇ means the difference, which are also common in time series. We have marked the answers to the questions in line 288.

 The meanings of the letters in Equation (4) are the same as the previous equations. We have marked these meanings in the corresponding sections of the manuscript. We have marked the answers to the questions in line 181.

Other errors have been corrected according to your comments. Both the figure and the table have been revised according to your comments, and the detailed changes are shown in the highlighted section of the revised draft.

Finally, thank you again for your comments to us.

Round 2

Reviewer 1 Report

Although the authors have made some changes, I still believe that this study has major methodological flaws.

Author Response

Thank you for your suggestions. We will try to do better in the future.

Reviewer 2 Report

Thank you for rectifying the manuscript based on the reviewers' comments 

Author Response

Thank you for your efforts and contributions to this manuscript, which has been much improved by your suggestions.

Reviewer 3 Report

First of all, I advise the authors in the future to answer each comment of the reviewer, not just a selection of comments which they believed were more relevant. I appreciate the authors tried to address each of my comments. However, some of them still require attention. 

The phrase at lines 69-72 now makes sense. However, it still has no predicate. Please correct.

The phrase at lines 77-79 has to be made clearer. When I read the previous sentence, I expect to see some examples for remote sensing application for soil moisture prediction. This is the case with no (1) , but not with no (2). I think the authors want to present the advantages (1) and disadvantages (2) of RS . So please reformulate.

line 106 - the potential evapotranspiration value of 1664 mm seems to high given the low annual temperature (0.9). Please check.

In the 2.1 section, I still don't see any information regarding the soil characteristics, which are very relevant in a study of soil moisture. The authors should provide this information (soil texture, structure, bedrock). If you don't have it in the original data sources, at least try to infer it from a soil map of Mongolia, or a digital soil map of the world like soilgrids.

Please try to use a World Reference Base name of the soil instead of Chesnut soil 

Figure 1 is not suggestive. It should present the location of the study area within Mongolia. It would be better to show elevation classes, main rivers and towns.

lines 124-127 - I see the authors have replaced the initial data sources links with a single link leading to a mathematical contest. This is confusing to me. What is the connection between these sources and this contest? I think it is better to keep the original links but update them

line 131 - the authors replaced "cyclic" with "goes round and round" which is a worse choice. I suggest the following formulation: the water variation ... is cyclic

lines 166-172 - I appreciate the authors have explained the parameters, but I also want to know what the abbreviations stand for (eg. AIC (Akaike information criterion)

line 205 - you don't need to insert a new section, especially since it only has a paragraph of text. I suggest you move the figure and the respective text just after "2.2 Methods" section

line 249 - why didn't you deleted "in the text" . The formulation "linear regression ... were constructed in the text" doesn't make sense to me. 

Author Response

Thank you for your comments again, we have made changes to the comments you made. The specific changes are as follows.

Q1: The phrase at lines 69-72 now makes sense. However, it still has no predicate. Please correct.

A1: We thank the reviewer for pointing out our error, which we have reorganized the language. We have marked the answers to the questions in line 69.

Q2: The phrase at lines 77-79 has to be made clearer. When I read the previous sentence, I expect to see some examples for remote sensing application for soil moisture prediction. This is the case with no (1) , but not with no (2). I think the authors want to present the advantages (1) and disadvantages (2) of RS . So please reformulate.

A2: We thank the reviewer for pointing out our error, which we have reorganized the language. We have marked the answers to the questions in line 77.

Q3: line 106 - the potential evapotranspiration value of 1664 mm seems to high given the low annual temperature (0.9). Please check.

A3: We have double-checked that the data is correct. The annual potential evaporation of Xilingol League ranges from 1500-2700mm (https://baike.baidu.com/item/%E9%94%A1%E6%9E%97%E9%83%AD%E5%8B%92%E7%9B%9F/5020479?fromModule=search-result_lemma). We have marked the answers to the questions in line 107.

Q4: In the 2.1 section, I still don't see any information regarding the soil characteristics, which are very relevant in a study of soil moisture. The authors should provide this information (soil texture, structure, bedrock). If you don't have it in the original data sources, at least try to infer it from a soil map of Mongolia, or a digital soil map of the world like soilgrids.

A4: Thanks for the reviewer's suggestion. We added the following soil characteristics. The soil is classified as kastanozem based on the World Reference Base for Soil Resources with 37% clay content (<0.002 mm), 42% silt content (0.002–0.02 mm), and 21% sand content (0.02–2 mm). The profile construct is Ah-Bk-C type. The organic matter content of this soil is 1.2%, soil pH—7.3, cation exchange capacity—22 cmol/kg, base saturation percent—100%, calcium carbonate content − 6%. We have marked the answers to the questions in line 123.

Q5: Please try to use a World Reference Base name of the soil instead of Chesnut soil

A5: Thanks for the reviewer's suggestion. In the World Reference Base we found Kastanozems to replace Chesnut soil. We have marked the answers to the questions in line 109.

Q6: Figure 1 is not suggestive. It should present the location of the study area within Mongolia. It would be better to show elevation classes, main rivers and towns.

A6: Thanks for the reviewer's suggestion. We have added its specific location in Inner Mongolia. In addition, we added terrain to the map. We have marked the answers to the questions in line 121.

Q7: lines 124-127 - I see the authors have replaced the initial data sources links with a single link leading to a mathematical contest. This is confusing to me. What is the connection between these sources and this contest? I think it is better to keep the original links but update them

A7: Thanks for the reviewer's suggestion. Now we have replaced the original defunct website. The data actually comes from this contest, and the documentation for this contest says that the sources of the data are these websites. We will later upload all the raw data to the system for easy access by readers. We have marked the answers to the questions in line 129.

Q8: line 131 - the authors replaced "cyclic" with "goes round and round" which is a worse choice. I suggest the following formulation: the water variation ... is cyclic

A8: We thank the reviewer for pointing out our error, which we have reorganized the language. We have marked the answers to the questions in line 148.

Q9: lines 166-172 - I appreciate the authors have explained the parameters, but I also want to know what the abbreviations stand for (eg. AIC (Akaike information criterion)

A9: Thanks for the reviewer's suggestion. We have labeled the full name of the abbreviation. We have marked the answers to the questions in line 184.

Q10: line 205 - you don't need to insert a new section, especially since it only has a paragraph of text. I suggest you move the figure and the respective text just after "2.2 Methods" section

A10: Thanks for the reviewer's suggestion. We have moved it after "2.2 Methods" section. We have marked the answers to the questions in line 139.

Q11: line 249 - why didn't you deleted "in the text". The formulation "linear regression ... were constructed in the text" doesn't make sense to me.

A11: Thanks for the reviewer's suggestion. We have removed "in the text". We have marked the answers to the questions in line 264.

Thank you for your efforts and contributions to this manuscript. Your suggestions have improved it a lot.